# Recent Advances in Equalization Technologies for Short-Reach Optical Links Based on PAM4 Modulation: A Review

**Honghang Zhou, Yan Li \*, Yuyang Liu, Lei Yue, Chao Gao, Wei Li, Jifang Qiu, Hongxiang Guo, Xiaobin Hong, Yong Zuo and Jian Wu \***

The State Key Laboratory of Information Photonics and Optical Communications, Beijing University of Posts and Telecommunications, Beijing 100876, China; zhh1994@bupt.edu.cn (H.Z.); yuyangliu@bupt.edu.cn (Y.L.); leiyuebupt@outlook.com (L.Y.); gcg@bupt.edu.cn (C.G.); w_li@bupt.edu.cn (W.L.); jifangqiu@bupt.edu.cn (J.Q.); hxguo@bupt.edu.cn (H.G.); xbhong@bupt.edu.cn (X.H.); yong_zuo@bupt.edu.cn (Y.Z.)

\* Correspondence: liyan1980@bupt.edu.cn (Y.L.); jianwu@bupt.edu.cn (J.W.)

**Abstract:** In recent years, short-reach optical links have attracted much more attention and have come to constitute a key market segment due to the rapid development of data-center applications and cloud services. Four-level pulse amplitude modulation (PAM4) is a promising modulation format to provide both a high data rate and relatively low cost for short-reach optical links. However, the direct detector and low-cost components also pose immense challenges, which are unforeseen in coherent transmission. To compensate for the impairments and to truly meet data rate requirements in a cost-effective manner, various digital signal processing (DSP) technologies have been proposed and investigated for short-reach PAM4 optical links. In this paper, an overview of the latest progress on DSP equalization technologies is provided for short-reach optical links based on PAM4 modulation. We not only introduce the configuration and challenges of the transmission system, but also cover the principles and performance of different equalizers and some improved methods. Moreover, machine learning algorithms are discussed as well to mitigate the nonlinear distortion for next-generation short-reach PAM4 links. Finally, a summary of various equalization technologies is illustrated and our perspective for the future trend is given.

**Keywords:** short-reach optical links; direct detection; four-level pulse amplitude modulation; digital signal processing; equalization

---

## 1. Introduction

Driven by upcoming services such as the Internet of Things (IoT), 4K/8K video applications, virtual reality (VR) and big data, the global internet protocol (IP) traffic has grown explosively in recent years [1]. As predicted by a Cisco report [2], 4.8 zettabytes of annual global IP traffic will be reached by 2022 and the so-called "Zettabyte Era" has arrived. To accommodate the associated demands, short-reach optical links have been widely investigated for data center interconnects (DCI), metro network, optical access, and so forth [3]. Unlike long-reach transmission, the short-reach optical links are especially sensitive to cost and size due to the large scale of their deployment [4]. Therefore, intensity modulation with direct detection (IM/DD) is adopted as the mainstream solution instead of coherent detection [5–7]. Traditional IM/DD optical interconnects implemented with non-return-to-zero on-off-keying (NRZ-OOK) format struggle to support the requirements of the increased transmission rate. Thus, many advanced modulation formats are employed to improve the spectrum efficiency (SE) and reduce the bandwidth limitation for electronic and optical components [8–17], such as four-level pulse amplitude modulation (PAM4), carrier-less amplitude and phase modulation (CAP), discrete

multi-tone (DMT), and quadrature amplitude modulation (QAM) based on Kramers-Kronig (KK) receiver [18]. Considering the power consumption and implementation complexity, the most attractive format in short-reach optical links is PAM4, which is believed as a highly promising candidate for the next-generation passive optical network (NG-PON) and has been ratified by IEEE 802.3 bs for 400 Gbps Ethernet transmission [19–25].

However, the low-pass filtering effects induced by the limited bandwidth of transmitter and receiver can cause severely inter-symbol interference (ISI). Low-cost devices such as lasers, modulators, photodiodes (PD) and trans-impedance amplifiers (TIA) also produce nonlinear distortions like level-dependent skew and level-dependent noise. Furthermore, the interaction between chromatic dispersion (CD) and direct detection will lead to a power-fading effect, where the detected signal may contain frequency notches after several kilometers transmission at a high symbol rate. Therefore, various equalization technologies based on digital signal processing (DSP) have been investigated for short-reach links over the years [26–53]. Conventional equalizers, such as the feed-forward equalizer (FFE) or decision feedback equalizer (DFE), are employed to compensate for the linear impairments induced by limited bandwidth and CD [26–28], while the equalizer based on the Volterra series is utilized to mitigate nonlinear distortion [29–32]. Some new equalization techniques, such as direct detection-faster than Nyquist (DD-FTN) algorithm [33–35], intensity directed FFE/DFE (ID-FFE/ID-DFE) algorithm [36,37] and joint clock recovery and FFE (CR-FFE) algorithm [38,39], have recently been proposed to solve different problems in practical implementation. In addition, machine learning methods like support vector machine (SVM) [40–42] and neural network (NN) [43–53] have been investigated to further eliminate system impairments for PAM4 modulation.

In this paper, an overview is provided on the important topic of advanced DSP for short-reach PAM4 optical links. The rest of this paper is organized as follows. In Section 2, the system configuration is introduced for short-reach PAM4 transmission in detail, where the comparisons for different transmitters and receivers are presented as well. Section 3 describes the distortions of high-speed transmission with low cost, including limited bandwidth, CD, nonlinear devices and power fading effect. To deal with these impairments in practical implementation, the principles of conventional FFE/DFE and improved algorithms based on FFE/DFE are first illustrated in Section 4. Moreover, VNLE and equalization based on machine learning are also discussed here. Furthermore, a summary of various equalization technologies is given in Section 4. Finally, conclusions are drawn in Section 5.

## 2. System Configuration

The typical system configuration for short-reach optical links based on PAM4 modulation is presented in Figure 1. The origin data generated by pseudorandom binary sequence (PRBS) is first encoded by a forward error correction (FEC) encoder and then mapped to the PAM4 symbol. After resampling and pulse shaping, the digital signal is pre-equalized through a finite impulse response (FIR) filter to pre-compensate for the limited bandwidth and nonlinearity of the transmitter. The processed signal is then loaded into a digital-to-analog converter (DAC) in order to obtain the baseband electrical signal. Afterward, the DAC output signals are amplified by a linear driver and then fed into a modulator for the generation of optical PAM4 signal. The laser can be a vertical cavity surface-emitting laser (VCSEL), a directly modulated laser (DML) or an electro-absorption modulator integrated with distributed feedback laser (EML). A comparison between different kinds of transmitter is summarized in Table 1. Generally, the directly-modulated lasers like VCSEL and DML reach a bandwidth barrier to operate beyond 25 GBaud [54], despite many research efforts made to achieve higher speed transmission [55,56]. EML generally have high bandwidth at the expense of high cost. Since the electrical signals of VCSEL and DML are directly applied to their laser cavity, larger frequency chirps will exist rather than EML, which can induce severe nonlinear distortions and lead to the serious degradation of transmission performance [57,58]. However, the adiabatic chirp of DML causes frequency modulation (FM) of the optical signal, which is able to fundamentally mitigate the first power fading dip incurred by CD [59]. Note that the Mach-Zehnder modulator (MZM) based on

Lithium Niobate (LiNbO$_3$) is out of consideration due to a large footprint and high cost. For optical links below 300 m, the multi-mode fiber (MMF) combined with VSCEL is widely used at present. When the transmission distance increases, the modal dispersion in the MMF will distort the signal and the standard single-mode fiber (SSMF) becomes the common choice.

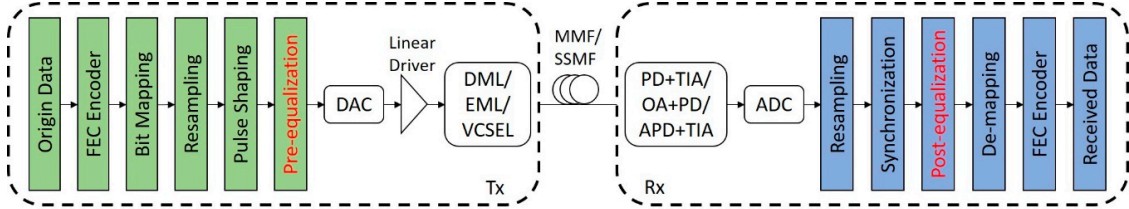

**Figure 1.** Typical configuration for short-reach PAM4 transmission system.

**Table 1.** Comparison of different transmitters [54–59].

| Tx | Bandwidth | Chirp | Cost | Reach | Fiber | Power Fading | Wavelength (nm) |
|------|-----------|-------|------|--------|-------|--------------|-----------------|
| VSCEL | low | high | low | <300 m | MMF | N/A | Mostly 850 |
| DML | low | high | fair | <80 km | SSMF | low | 1310/1550 |
| EML | high | low | high | <80 km | SSMF | high | 1310/1550 |

At the receiver, the signal is directly detected by PD or avalanche photodiode (APD). To improve receiver sensitivity, a TIA is generally cascaded behind to amplify the electrical signal. An alternative solution is using the combination of an optical amplifier (OA) and an optical band-pass filter (OPBF) before PD to amplify received optical signal and remove out-of-band noise. The comparison of different receivers is presented in Table 2, where OA + PD has the highest sensitivity due to the use of high-cost OA. The gain of APD results in relative higher SNR, which makes it more suitable for longer reach links compared to traditional PD [26]. After being processed by an analog-to-digital converter (ADC), the signal is resampled to an expected sampling rate for subsequent steps. Synchronization is achieved to eliminate sampling clock offset (SCO) and depress timing jitter. Then, various DSP techniques are utilized to equalize the received signal for the performance improvement of the transmission system. Finally, after de-mapping and FEC decoding, the received data is obtained and the bit error rate (BER) can be calculated.

**Table 2.** Comparison of different receivers [1,7,26,39].

| Rx | Cost | Footprint | Sensitivity | Reach |
|----------|------|-----------|-------------|-------|
| PD + TIA | low | low | low | low |
| APD + TIA | fair | low | fair | fair |
| OA + PD | high | high | high | high |

However, due to the limitation of cost, all transmitters and receivers mentioned above can induce different degree distortions for high-speed PAM4 transmission. Therefore, equalization is one of the most DSP components to eliminate impairments in short-reach optical links. As is shown in Figure 1, the equalization technologies can be utilized both in the transmitter as pre-equalization and in the receiver as post-equalization. The distortions induced by low cost and the different equalization technologies will be introduced specifically in the following sections.

## 3. Distortions Induced by Low Cost

In short-reach optical links based on PAM4 modulation, the cost is one of the most important factors to be considered for commercial implementation. Low-cost components are highly desired in practical application, while they also bring great challenges and serious distortions such as linear and

nonlinear impairments. In this section, the major distortions induced by low cost will be illustrated in short-reach optical communications.

### 3.1. Linear Impairments: Limited Bandwidth and Chromatic Dispersion

The high-speed PAM4 system suffers from severe bandwidth limitation due to the consideration of low cost. After bandwidth limited devices, the received signal with low-pass filtering effects can be expressed as:

$$y_k = I_k + \sum_{\substack{n = 0 \\ n \neq k}}^{\infty} I_n x_{k-n} + v_k \tag{1}$$

where $x_k$ and $y_k$ is the input signal and output signal at the $k$-th sampling instant, respectively. The first term $I_k$ represents the desired information symbol while the second term donates the ISI. The last term $v_k$ is the additive Gaussian noise variable at the $k$-th sampling instant. To investigate the performance impact of bandwidth limitation, a typical measured system frequency response is shown in Figure 2a [60]. The 3 dB bandwidth of the total transmission system is approximately 7.5 GHz, which is far below the required bandwidth for 64 Gbaud PAM4 signal. The attenuation of amplitude frequency is about 15 dB at the Nyquist frequency. In the insets (I) of Figure 3a, it can be observed that the optical eye diagram is confusing, and even four levels of PAM4 signal are unable to be detected. After compensation, a great improvement is achieved in the quality of the eye diagram, which reveals the tremendous influence of limited bandwidth. As shown in the blue part of Figure 2b, the receiver sensitivity penalty is measured as a function of the bandwidth for PAM4 system in simulation [19]. The received optical power at BER = $3.8 \times 10^{-3}$ for a bandwidth of 35 GHz is employed as reference. It can be seen that approximately 3.3 dB received power is lost as the bandwidth decreases from 35 GHz to 15 GHz. Moreover, the receiver sensitivity penalty increases in speed when the bandwidth limitation becomes severe. The other simulation for PAM4 transmission is depicted in the black part of Figure 2b, where the bandwidths of the components are varied uniformly [61]. The total bit rate is fixed at 96 Gbps, while the 3 dB bandwidths of DAC, modulator, PD and ADC are changed from −30 to 30% simultaneously. The obtained BER performance is degraded as the bandwidth reduces, which confirms that the narrow bandwidth is one of the factors that deteriorate the performance of PAM4 optical links.

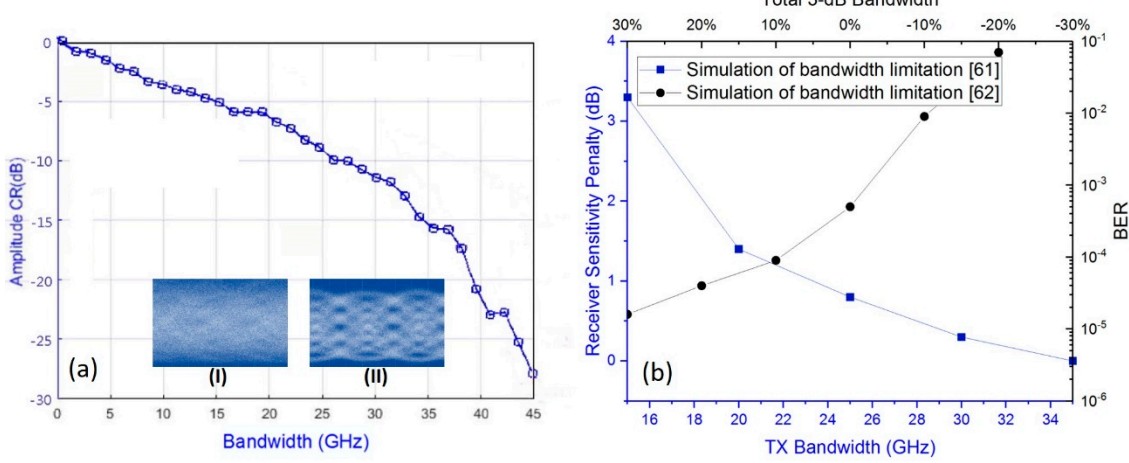

**Figure 2.** (**a**) Frequency response in band-limited system and eye diagrams (I) before and (II) after bandwidth compensation (redrawn after [60]). (**b**) Effect of bandwidth to system performance (redrawn after [19,61]).

When PAM4 is used in the C-band, the CD tolerance is considered as one key drawback, which critically impairs the high-frequency components of the signal. The typical tolerance to residual

CD in terms of the optical signal-to-noise ratio (OSNR) for PAM4 is shown in Figure 3, where the OSNR penalty is about 1 dB for dispersion within ±100 ps/nm [62]. Meanwhile, the eye diagrams for 56 Gbps PAM4 signal after 0 km and 20 km transmission are shown in Figure 3b,c, respectively. A qualitative comparison between these two eye diagrams reveals that CD brings serious degradation in signal quality.

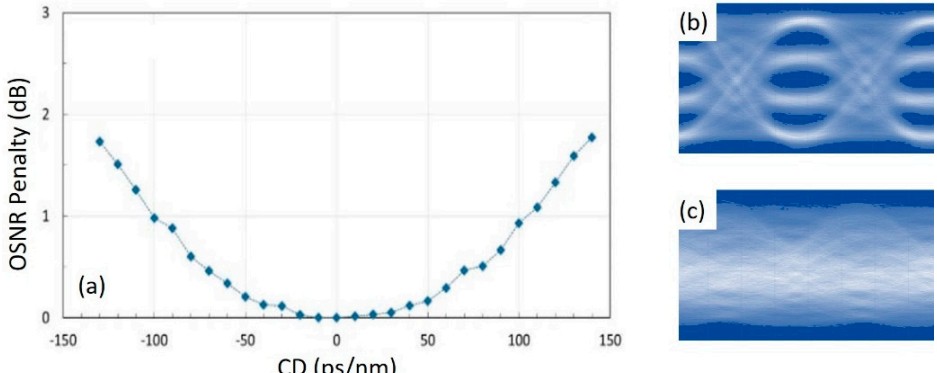

**Figure 3.** (**a**) Four-level pulse amplitude modulation (PAM4) signaling tolerance on dispersion in terms of optical signal-to-noise ratio (OSNR) penalty (reprinted from [62] with permission from authors). Eye diagrams for 56 Gbps PAM4 signal after (**b**) 0 km and (**c**) 20 km transmission.

### 3.2. Nonlinear Impairments

#### 3.2.1. Nonlinear Devices: Level-Dependent Noise and Level-Dependent Skew

The low cost of devices not only causes the bandwidth limitation but also produces nonlinear distortions like amplitude-dependent noise and level-dependent skew. An example for APD or PD with the optical amplifier is depicted in Figure 4, where different power levels of the PAM4 signal have different distributions of dominant noise [63]. The probability of distributions of regular equally-spaced PAM4 signal after OA + PD and APD receivers are illustrated in Figure 4a,b, respectively. The peaks of probability density represent different noise variances for different levels. From Figure 4c we can notice that high power levels suffer larger noise interference and more symbols cross the decision thresholds, which is consistent with the probability of distributions. Note that the distribution of noise depends on which impairment is dominant, so it is diversified. For instance, the noise of two middle levels could be larger than the noise of two marginal levels considering the nonlinearity of the modulation curve.

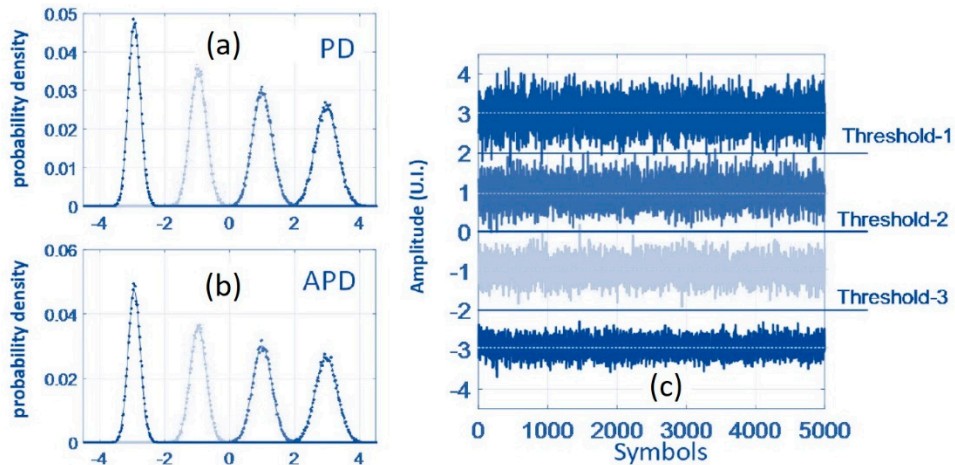

**Figure 4.** (**a**) The nonlinear probability density distributions of received PAM4 signal for (**a**) OA + PD, and (**b**) APD. (**c**) The waveform of received PAM4 signal after APD (reprinted from [63] with permission from authors).

The level-dependent skew induced by the nonlinear chirp characteristic of modulators can produce significant penalties due to the differences in optimum sampling time for the different level of the eye. As shown in Figure 5a,b, the simulated PAM4 signal without any skew and the simulated 50 Gb/s PAM4 signal using spatial-temporal rate equation modeling [64]. An obvious amplitude-dependent eye can be observed, and hence this inevitably introduces timing impairments. The measured eye diagrams for different signal rates and bias currents of the laser are presented in Figure 5c–f. These figures indicate that the phenomenon of level-dependent skew is more serious under the condition of high-speed and low power consumption, which severely degrades the transmission performance.

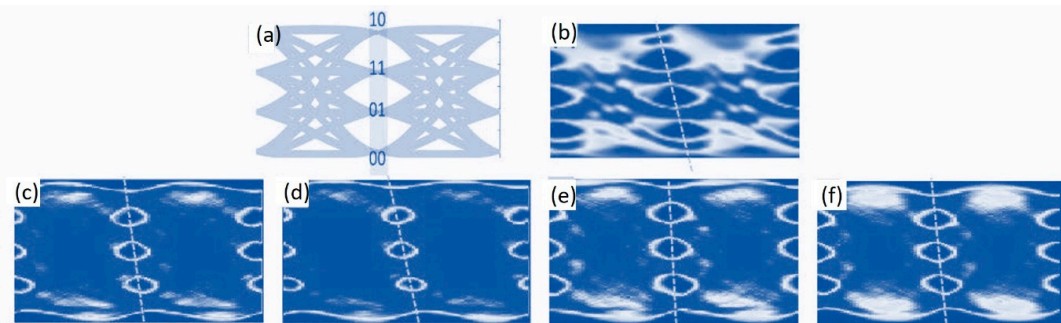

**Figure 5.** The eye diagrams of (**a**) ideal PAM4 signal; (**b**) simulated PAM4 at 50 Gb/s using spatial-temporal modeling, (**c**) measured 56 Gb/s PAM4 signal when bias current is 5 mA; (**d**) measured 64 Gb/s PAM4 signal when bias current is 5 mA; (**e**) measured 56 Gb/s PAM4 signal when bias current is 7 mA; (**f**) measured 64 Gb/s PAM4 signal when bias current is 7 mA (reprinted from [64] with permission from authors).

### 3.2.2. Power Fading Effect

Due to the interaction between chromatic CD and direct detection, the induced power-fading effect will produce spectral zeros in the spectrum of the signal, which is the key points to determine transmission distance. Figure 6a–c plots the simulated magnitude response of a 28 Gbaud PAM4 system for various fiber lengths, namely 15 km, 50 km, and 100 km, respectively [65]. It can be clearly seen that the number of spectral zeros changes from none to one and then to three by increasing the transmission distance. Meanwhile, the probability of frequency notches in the high frequency is greater, which proves that the power-fading effect is more severe for high-speed transmission. For experimental demonstrate, the frequency response of the received 25 Gbaud PAM4 signal over 50 km SSMF transmission is depicted in Figure 6d [66]. The first frequency notch can be obviously found around 9 GHz, which indicates that the power fading effect is an inescapable problem in IM/DD system.

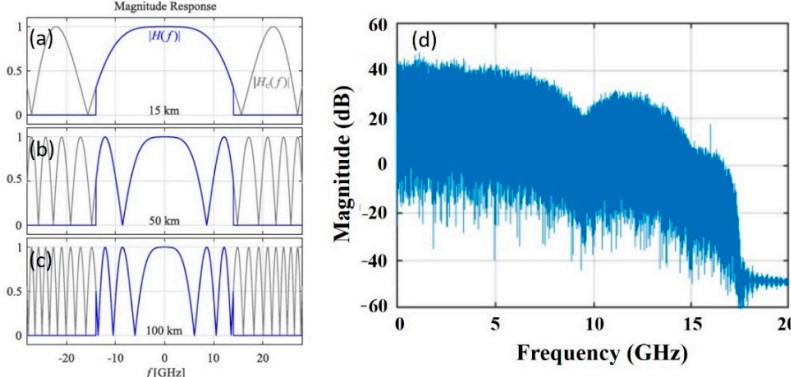

**Figure 6.** Magnitude responses of a 28 Gbaud PAM4 system after (**a**) 15 km; (**b**) 50 km; (**c**) 100 km in simulation (reprinted from [65] with permission from IEEE). (**d**) Frequency spectrum of the detected 25 Gbaud PAM4 signal over 50 km SSMF transmission (reprinted from [66] with permission from authors).

## 4. Equalization Technologies

To mitigate the impairments in high-speed PAM4 system mentioned above, various digital equalization technologies have been studied. In this section, we introduce the most popular equalizers, including FFE, DFE, Volterra nonlinear equalizer (VNLE) and machine learning based equalizer, in detail to compensate for these impairments. Moreover, some improved equalization methods are also described to handle specific issues here.

### 4.1. FFE/DFE

### 4.1.1. Conventional FFE/DFE

The feed-forward equalizer is an effective method for linear impairments compensation and widely used nowadays. The most basic components of FFE is the finite impulse response (FIR) filter, whose structure is shown in Figure 7a. The output of FIR is expressed as [67]

$$y(k) = \sum_{l=0}^{n-1} h_l x(k-l) \tag{2}$$

where $x(k)$ and $y(k)$ are the input and output signal of FIR at the sampling instant $k$, respectively. $h = [h_0 h_1 h_2 \ldots h_{n-1}]$ is the array of tap weights, while $n$ is the number of taps.

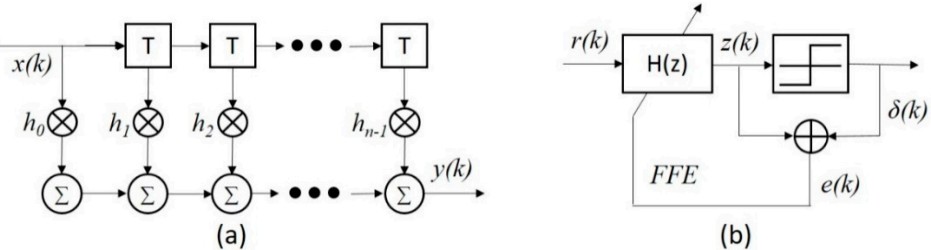

**Figure 7.** The structure of (**a**) FIR filter: H(z); (**b**) FFE.

Figure 7b depicts the structure of decision-directed FFE, where the FIR filter is noted as $H(z)$. The tap weights can be updated by the zero-forcing (ZF) algorithm, least mean squares (LMS) algorithm, recursive least squares (RLS) algorithm and so on [67–69]. Different convergence algorithms just affect the speed of obtaining optimal tap weights and are not the focus of this review. Here, we only illustrate one of the most common techniques called direct decision LMS (DD-LMS) algorithm. The $(k+1)$-th update of the filter tap weights is given by [68]

$$h(k+1) = h(k) + 2\mu e(k)R(k) \tag{3}$$

where $\mu$ is the step size and the error $e(k) = \delta(k) - z(k)$ is between the desired signal $\delta(k)$ and the output signal $z(k)$. The $R(k) = [r(k), r(k-1), r(k-2), \ldots, r(k-n+1)]$ is the vector of the input signal. The FFE can boost the power of high-frequency components that undergo large losses due to the system bandwidth limitations. It should be noted that FFE can operate at symbol rate sampling or higher. Compared with symbol-spaced FFE, the fractionally-spaced FFE which is sampled at several times the symbol rate allows the matched filter to be realized digitally and can take care of phase recovery. In addition, it has a lower residual error at the cost of computational complexity. The destructive effect of the frequency notches is unable to be compensated for easily with an FFE but could be efficiently mitigated using a DFE. Unlike FFE, the input of DFE is the signal after decision, as shown in Figure 8a.

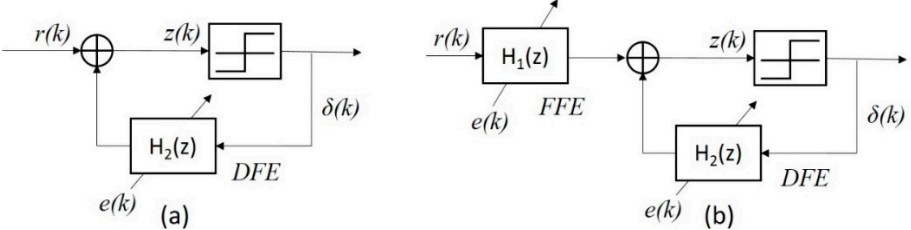

**Figure 8.** The structure of (**a**) DFE; (**b**) FFE and DFE combined.

In the case of DFE, the signal before decision is calculated by [69]

$$z(k) = r(k) - \sum_{l=0}^{n-1} h_l \delta(k-l) \tag{4}$$

The $(k + 1)$th update of the filer tap weights is given by [69]

$$h(k+1) = h(k) + 2\mu e(k)Z(k) \tag{5}$$

where $Z(k) = [z(k), z(k-1), z(k-2), \ldots, z(k-n+1)]$. DFE is usually operated at one sample per symbol. It should be noted that DFE can successfully equalize frequency notches by pole insertion, but it may suffer from error propagation and is unstable due to the decision feedback scheme. Moreover, only post-cursor ISI can be deal with DFE, while the CD-induced channel impulse response contains both pre-cursor and post-cursor. Therefore, the best choice for practical implementation is a combination of an FFE and a DFE, which can be seen from Figure 8b. The FFE and DFE can be placed in the transmitter for pre-compensation or in the receiver for post-compensation [60]. To solve the problem of error propagation, a transmitter-side DFE called Tomlinson-Harashima pre-coding has recently been proposed and investigated in various PAM4 short-reach optical links [70–73]. In addition, many novel equalizers based on conventional FFE/DFE are proposed and investigated to deal with the problems in practical implementation and improve the transmission performance. Next, three recently proposed equalizers including DD-FTN, ID-FFE/ID-DFE, and CR-FFE will be illustrated as examples.

4.1.2. Improved Algorithms Based on FFE/DFE

● DD-FTN

When the FFE tries to amplify the power of high-frequency components to compensate for the low-pass effects, it boosts the noise power at high-frequency components as well. To address this phenomenon, a DD-FTN algorithm is proposed [33] and the principle is shown in Figure 9.

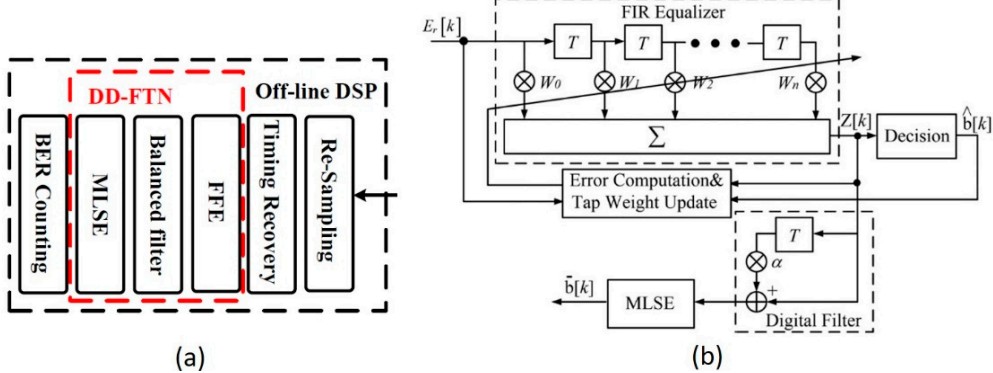

**Figure 9.** (**a**) The components and (**b**) the structure of DD-FTN (reprinted from [33] with permission from authors and reprinted from [7] with permission from IEEE).

First, the signal after synchronization is equalized by a FFE, adapted by the LMS algorithm. Then, a balanced digital filter is placed behind to suppress the enhanced in-band noise by the equalizer. The transfer function of the balanced filter in *z*-transform is $H(z) = 1 + \alpha z^{-1}$, where the tap coefficient $\alpha$ is employed to optimize the frequency response of the post filter. Finally, the maximum likelihood sequence estimation (MLSE) is utilized behind to eliminate the strong ISI induced by the balanced filter. Although two more DSP blocks are contained in DD-FTN compared to traditional equalizers, the increase in complexity is relatively small and the overall complexity is not high compared to other DSP techniques.

The BER performance as a function of received optical power for 112 Gbps PAM4 signal after 2 km SSMF transmission is depicted in Figure 10a [33]. With conventional FFE, the enhanced noise significantly degrades the system performance and a BER floor at $5 \times 10^{-2}$ can be observed. By using DD-FTN, the noise enhancement is effectively mitigated by the balanced filter and the induced ISI can be processed by the MLSE. The performance is improved and the BER can reach $3 \times 10^{-4}$ when received power is −7.1 dBm. Figure 10b shows the BER performance versus received optical power for 140 Gbit/s PAM-4 system after back-to-back transmission [35]. It can be found that the traditional FFE updated by DD-LMS exhibits the worst performance, while the BER can reach the hard decision FEC (HD-FEC) threshold of $3.8 \times 10^{-3}$ with DD-FTN. Therefore, DD-FTN can make a big difference in enhanced noise mitigation.

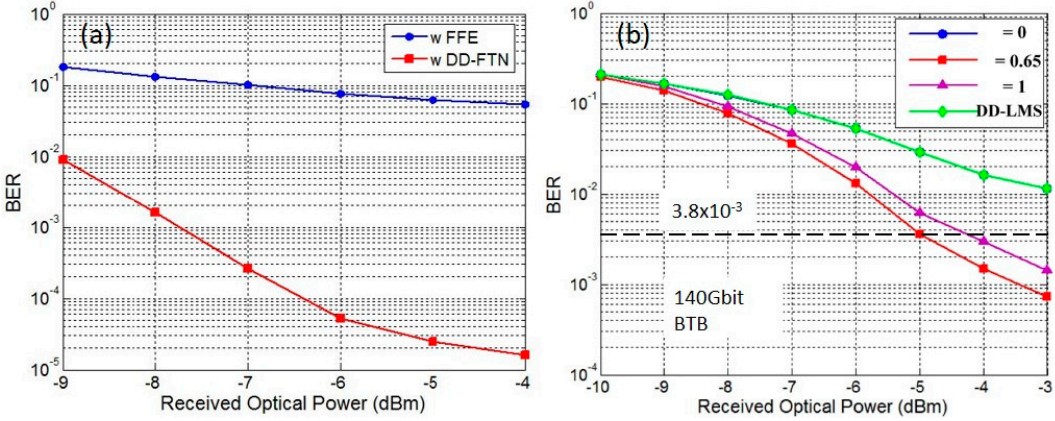

**Figure 10.** (**a**) The bit error rate (BER) performance versus received optical power for 112 Gbps PAM4 signal after 2 km SSMF transmission (reprinted from [33] with permission from authors). (**b**) The BER performance versus received optical power for different tap coefficient of balanced filter for 140 Gbit/s PAM4 system after back-to-back transmission (reprinted from [35] with permission from IEEE).

- ID-FFE/ID-DFE

In C-band transmission, low-cost modulator like DML will introduce an additional impairment due to the interaction between frequency chirp and chromatic dispersion. Recently, a low complexity intensity directed equalizer based on FFE and DFE is proposed to suppress the chirp induced distortions. Different from the traditional FFE/DFE that utilizes one set of coefficients, the proposed algorithm first divides the symbol into different sets according to its intensity level, as shown in Figure 11a [37]. Then the coefficients of different groups are applied accordingly. Four sets of coefficients for PAM4 symbol are selected according to the three thresholds for ID-FFE and four levels for ID-DFE. After classification, the desired tap coefficients from different sets are used and the following procedures are the same as the traditional FFE/DFE. In terms of complexity, only a few simple decision circuits are added to the original FFE/DFE and the complexity increment is negligible considering the operations of multiplication in the equalizer.

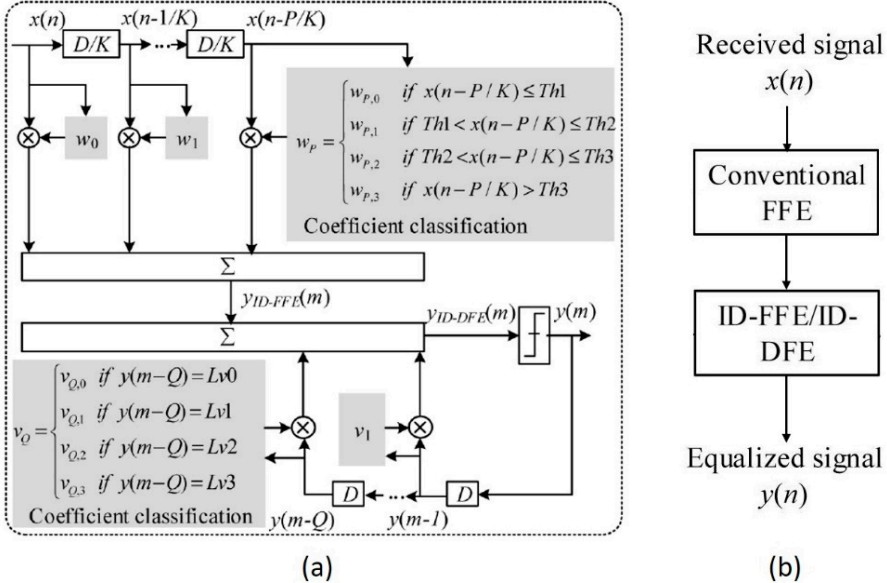

**Figure 11.** (**a**) The detailed structure of (P-1)-tap ID-FFE and Q-tap ID-DFE. (**b**) The block diagram of the pre-FFE + ID-FFE/ID-DFE (reprinted from [37] with open access from OSA).

The effectiveness of ID-FFE/ID-DFE greatly depends on the accuracy of the decision in the classification. Thus, a conventional FFE is employed to equalize the severe impairments and enhance the decision accuracy before the intensity directed equalizer as shown in Figure 11b.

The BER performance versus the received optical power is shown in Figure 12 [37]. Compared with the FFE case, of which the BER is $9 \times 10^{-3}$, the ID-FFE/ID-DFE reaches $2 \times 10^{-3}$. When the pre-FFE is used before ID-FFE/ID-DFE, the performance is improved by almost an order of magnitude and the BER reduce to $2.6 \times 10^{-4}$. From the received eye diagrams of ID-FFE with and without pre-FFE, we observe that the pre-FFE can completely suppress the residual eye skewing effect of the ID-FFE and dramatically remove the ISI. When the transmission distance increase to 43 km, only the ID-FFE/ID-DFE assisted by pre-FFE can reach a BER of $3.6 \times 10^{-3}$ below the HD-FEC. Therefore, the chirp of the modulator can be well addressed by the novel proposed ID-FFE/ID-DFE.

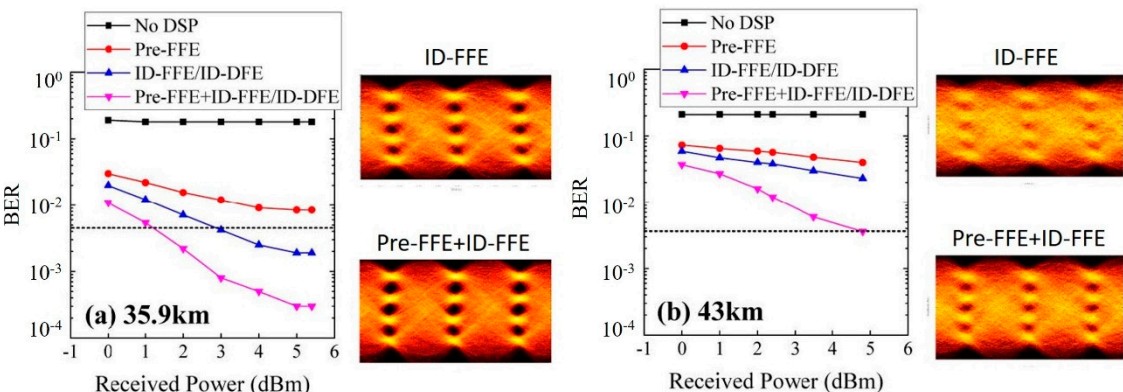

**Figure 12.** The BER performance of 56 Gbps PAM4 signal (**a**) after 35.9 km transmission and (**b**) after 43 km transmission (reprinted from [37] with open access from OSA).

- CR-FFE

The problem of incompatible prerequisites between impairment equalization and clock recovery also reduces the performance of PAM4 transmission [74]. A joint clock recovery and feed-forward equalization algorithm is proposed recently, which estimates timing error based on the difference

between two tap coefficients of T/2-spaced FFE. The proposed algorithm can eliminate the ISI induced by linear impairments and track large sampling clock offset (SCO) simultaneously.

The structure of the CR-FFE is plotted in Figure 13, where we notice that the timing error is derived from the difference between two tap coefficients [39]. For the comparison of complexity, the timing error of CR-FFE is calculated by one time subtraction, while the conventional CR algorithm needs an additional multiplication. Considering the total multiplication operations, the computational complexity of CR-FFE is similar to or slightly lower than that of the original scheme.

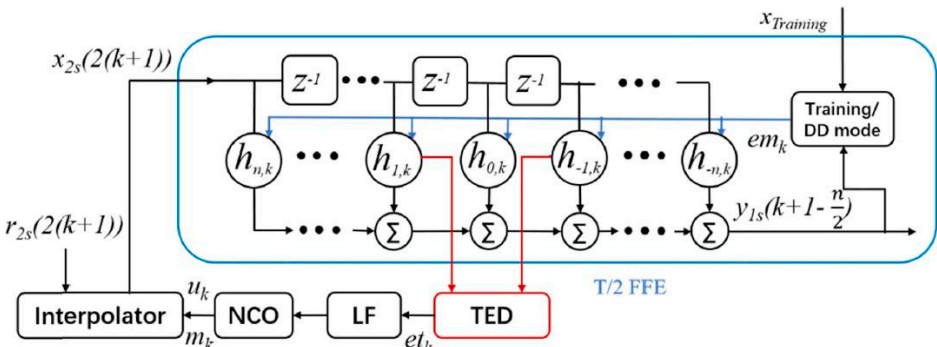

**Figure 13.** The structure of proposed joint clock recovery and FFE (CR-FFE) (reprinted from [39] with open access from OSA).

The BER performance of CR-FFE over different SCO is studied as shown in Figure 14a [39]. The traditional clock recovery cascaded by FFE (noted as scheme II) cannot counteract large SCO, while the CR-FFE (noted as scheme III) can ensure stable and reliable performance as SCO increase from 0 to 1000 ppm. From Figure 14b,c, we observe that the two tap coefficients for timing error detection are basically equal and the fractional interval has a stable changing process, which indicates that clock recovery and equalization are accomplished simultaneously after 40 km transmission when SCO reaches 1000 ppm. Therefore, the CR-FFE can solve the problem of incompatible prerequisites between impairment equalization and clock recovery, and provide significant system performance improvement for PAM4 transmission.

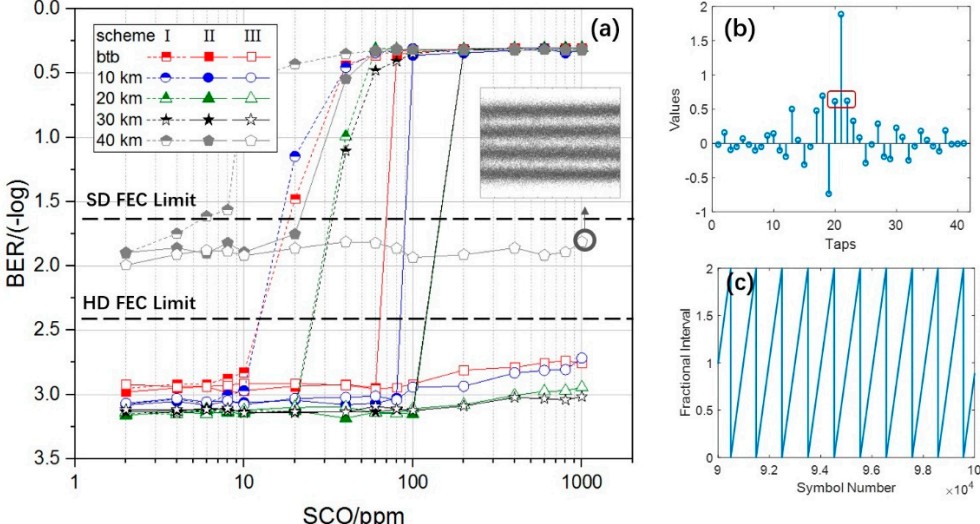

**Figure 14.** (**a**) The BER performance vs. sampling clock offset at received optical power of −8 dBm for 50 Gbps PAM4 signal; (**b**) the tap weights and (**c**) fractional interval of proposed CR-FFE after 40 km transmission when sampling clock offset is 1000 ppm (reprinted from [39] with open access from OSA).

*4.2. VNLE*

Thanks to FFE and DFE, the linear impairments can be efficiently eliminated. However, the residual nonlinear distortion mainly induced by the nonlinearity of devices and square-law detection can also severely impact the transmission performance of the PAM4 system. One of the most popular equalization techniques is VNLE, whose structure is shown in Figure 15. Note that the VNLE can be implemented based on FFE or DFE and we focus on FFE based VNLE in this paper. Considering the exponential growth of computational complexity, the third-order VNLE is introduced here.

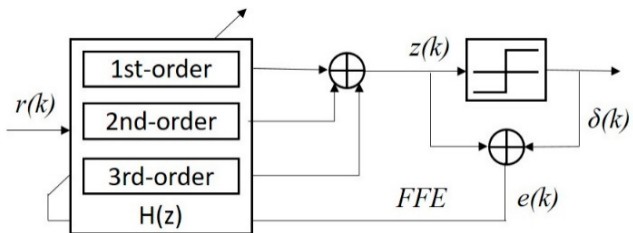

**Figure 15.** The structure of Volterra nonlinear equalizer (VNLE).

The output of VNLE is expressed as [75]

$$z(k) = \sum_{l=0}^{n_1-1} h_l x(k-l) + \sum_{l=0}^{n_2-1}\sum_{i=0}^{l} h_{l,i} x(k-l)x(k-i) + \sum_{l=0}^{n_3-1}\sum_{i=0}^{l}\sum_{j=0}^{i} h_{l,i,j} x(k-l)x(k-i)x(k-j) \quad (6)$$

where $h_l$, $h_{l,i}$ and $h_{l,i,j}$ are the tap weights of 1st-order, 2nd-order and 3rd-order kernels, respectively. $x(k)$ is the input signal of FIR in VNLE at the sampling instant $k$. In addition, $n_1$, $n_2$ and $n_3$ respectively represent the number of taps for the linear part, 2nd-order nonlinear part and 3rd-order nonlinear part. Generally, the linear impairments of the PAM4 system and the self-phase modulation (SPM) of SSMF can be successfully eliminated by the first and third-order kernels of VF, while the second-order kernels are utilized to compensate the nonlinearity of devices such as modulator and PD. The kernel coefficients are commonly updated by LMS algorithm considering complexity. To simplify the convergence procedure, the kernels of each order are evolved at different speeds [75], which can be depicted as a gradient vector $\mu = [\mu_1, \mu_2, \mu_3]$. However, there is no clear procedure on how to set the values of this vector.

While limited by the high computation complexity of fully-connected VNLE, it is not practical to be implemented directly in the real applications. Various methods have been proposed to reduce the computational complexity of VNLE and the interested readers can refer to [76–80] for a detailed discussion on this topic. Taking modified Gram-Schmidt orthogonal decomposition as an example, the performance of sparse VNLE is described as shown in Figure 16 [31]. The dependence of the BER on the back-to-back system without VNLE, and with VNLE or sparse VNLE are compared. As is shown, the required ROP at the FEC threshold is decreased by 0.7 dB for the VNLE. When sparse VNLE is employed, the increase in the ROP is less than 0.2 dB compared to VNLE. Figure 6b plots the dependence of the BER on the ROP for 10 km and 20 km transmission. No penalties are observed relative to the back-to-back system. Therefore, the sparse VNLE can maintain basically the same performance with a reduction of computational complexity, which is an excellent optimization for VNLE.

*4.3. Equalization Based on Machine Learning*

Machine learning (ML) is the scientific study of algorithms and statistical models that computer systems use to effectively perform a specific task without using explicit instructions, which was coined in 1959 by Arthur Samuel [81]. Recently, machine learning algorithms have been utilized to process optical communications and achieve distinguished performance [82–86]. Some DSP techniques based on machine learning including SVM and NN are described in this section for PAM4 optical links.

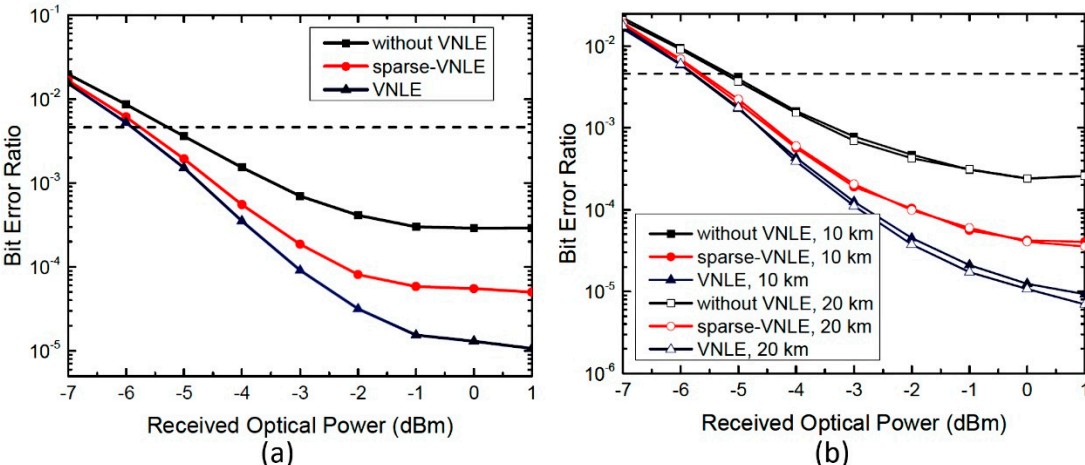

**Figure 16.** (**a**) BER vs. received optical power for a back-to-back system. (**b**) BER vs. received optical power for 10 km and 20 km transmission (reprinted from [31] with permission from authors).

### 4.3.1. Support Vector Machine

The basic SVM classifier is a two-class classifier and its training process can be described as finding the maximum edge hyperplane to define the decision function of the classifier. The concept of basic SVM classifier is shown in Figure 17a, where the margin means the minimum distance of all samples to the hyper-plane [40]. To solve this convex quadratic programming of optimization target in SVM, the sequential minimal optimization (SMO) algorithm [87] can be used.

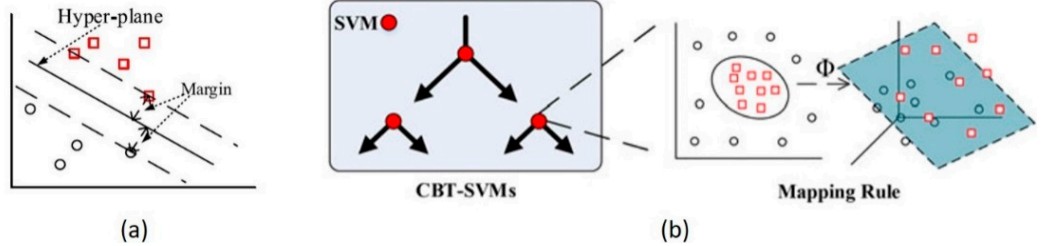

**Figure 17.** (**a**) Basic SVM classifier. (**b**) Complete binary tree structure multi-classes SVMs and mapping rule for PAM4 signal where the black circle and the red square represent two categories (reprinted from [40] with permission from IEEE).

Note that PAM4 cannot be de detected directly by the basic SVM classifier. Thus, a proposed complete binary tree (CBT) structure multi-classes SVMs method is utilized as depicted in Figure 17b. It can be seen that two layers are contained in the tree, where the first layer classifies the first bit of PAM4 signal, and the second layer classifies the last bit. Although the wrong decision in the high layer will be retained to the next layers, a significant improvement in performance for PAM4 system can still be expected.

The BER curves of the 40 Gbps and 50 Gbps PAM4 signal are shown in Figure 18a,b, respectively [42]. Using the proposed CBT-SVMs, significant receiver sensitivity reduction of 2.99 dB and 4.68 dB in optical B2B and 2 km SMF transmission can be obtained for 40 Gbps PAM4 system. Moreover, for 50 Gbps PAM4 modulation, it provides 3.59 dB and 3.63 dB received power tolerance compared with the hard decision. Therefore, the SVM algorithm shows outstanding performance in nonlinear impairment mitigation and could be a potential choice for future PAM4 optical links.

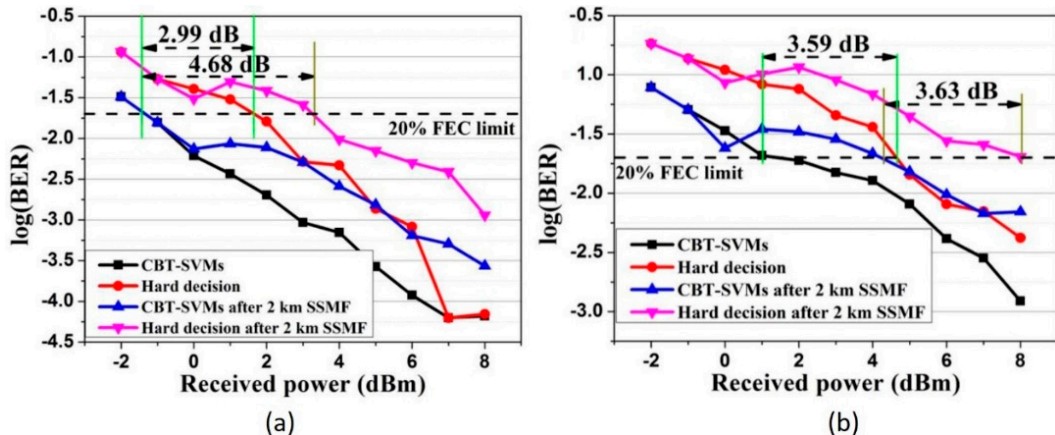

**Figure 18.** BER curves for (**a**) 40 Gbps and (**b**) 50 Gbps PAM4 signal using CBT-SVMs (reprinted from [42] with permission from IEEE).

### 4.3.2. Neural Network

The neural network (NN) is computational model loosely inspired by its biological counterparts [88]. In recent years, it has been proposed to mitigate the nonlinear impairments in optical communication system [89–91]. For short-reach PAM4 optical links, various research concerning the NN method has been performed to improve transmission performance [43–53]. The schematic of NN based nonlinear signal processing is presented in Figure 19a, where the leftmost part consists of a set of neurons representing the input features and the rightmost part is a non-linear activation function [48]. In the middle, a weighted linear summation is employed to represent connections between neurons. Figure 19b shows a two-layer neural network to classify PAM4 signals. Rectified Linear Units (ReLU), as indicated in the inset of Figure 19b, is always applied as the activation function. Compared with sigmoid function or Tanh function, it is more like a real neuron in our body and results in much faster training. A softmax function is commonly selected as the activation function for the output layer. The output of the softmax function is the class with the highest probability.

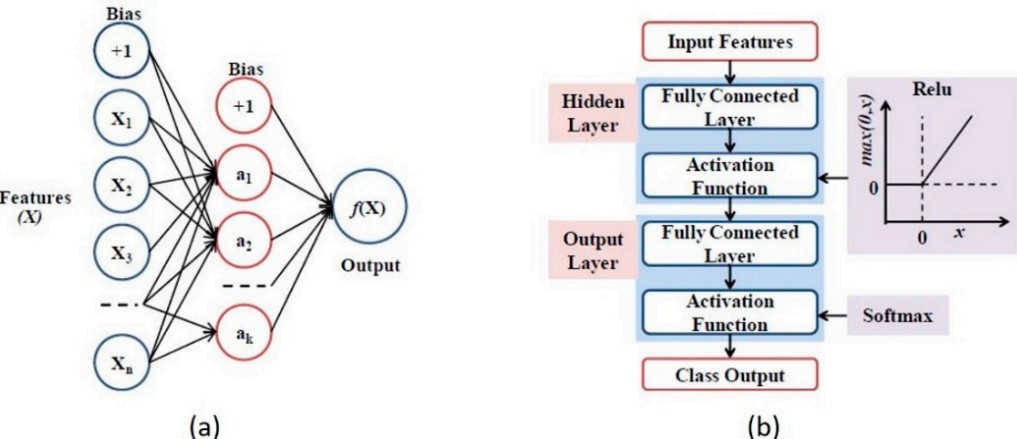

**Figure 19.** (**a**) Schematic of neural network (NN) based nonlinear signal processing for a hidden layer and (**b**) simple structure of two-layer neural network (reprinted from [48] with permission from authors).

An example for BER performance of NN is shown in Figure 20a [49]. Thanks to the NN method, about one order of magnitude is decrease compared to the BER with VNLE after 60 km transmission. Figure 20b shows the BER performance versus OSNR after 80 km SSMF. The required OSNRs to reach the FEC threshold for VNLE and NN are 39 dB and 35.5 dB, respectively. Thus, we can conclude that

compared to conventional equalizers, a better BER performance can be achieved using the NN method, which is an attractive solution for short-reach PAM4 optical links.

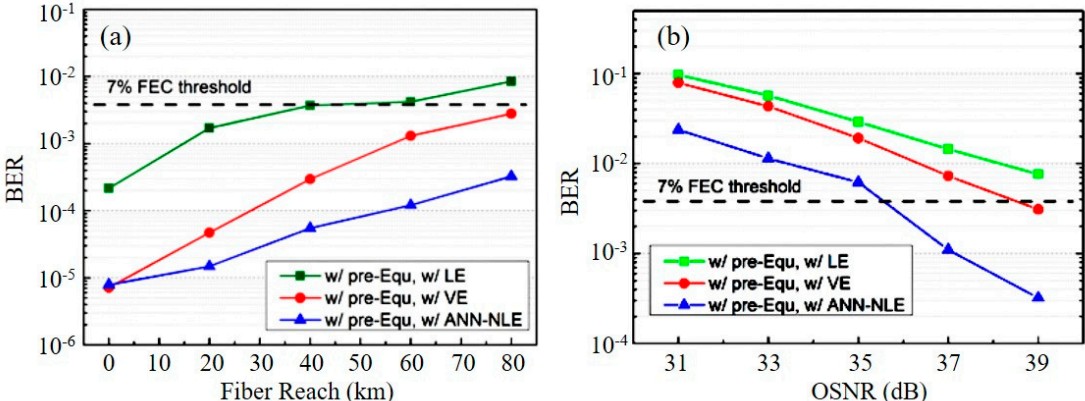

**Figure 20.** (**a**) BER vs. fiber length with NN for 112 Gbps SSB-PAM4 transmission. (**b**) BER vs. OSNR after 80 km dispersion uncompensated SSMF (reprinted from [49] with open access from OSA).

### 4.4. Summary of Recent Work

Various equalization technologies mentioned above are summarized and compared as shown in Table 3. The FFE/DFE is widely used in short-reach PAM4 optical links due to simple architecture and low complexity, however, it can only eliminate the linear ISI induced by bandwidth limitation and CD. In [26], a single-channel 56 Gbit/s PAM4 optical transmission over 60 km is experimentally demonstrated with receiver sensitivities of −19.9 dBm using 35 taps FFE. Meanwhile, with the help of 17 taps FFE, the 40 km error-free transmission is achieved for 106 Gbit/s PAM4 using APD receiver [27]. The THP is a transmitter-side DFE without suffering from error-propagation, which is proposed recently to resist against power fading and bandwidth limitation. For the first time, the THP is applied to 100 Gbit/s FTN PAM4 transmission over 40 km through a 20 GHz low-pass channel [72]. With similar or a low addition of complexity, the novel DD-FTN, ID-FFE/ID-DFE and CR-FFE can compensate for the weakness of conventional linear equalizers and achieve better performance for specific issues. In [35], the DD-FTN is experimentally demonstrated for a 140 Gbit/s PAM4 signal over 20 km transmission with a receiver sensitivity of −5.5 dBm. K. Zhang et al. firstly demonstrate a C-band 56 Gbit/s PAM4 system over 43 km transmission with the proposed ID-FFE/ID-DFE [37]. In [39], the CR-FFE resist SCO up to 1000-ppm after 40 km transmission for 50 Gbit PAM4 system based on 10 GHz DML. While bandwidth limitation can be efficiently compensated for by using FFE/DFE, the nonlinearity of devices is difficult to eliminate effectively. VNLE is the most popular nonlinear equalization while the high computational complexity can be reduced by removing the less important kernels. Using 1722 taps VNLE, the C-band 2 × 56 Gbit/s PAM4 transmission over 100 km is experimentally demonstrated in [92]. A sparse VNLE with half computational complexity achieves similar performance for 2×64 Gbit/s PAM4 transmission over 70 km SSMF using 18 GHz DML [93]. As for machine learning algorithms, SVM and NN including artificial NN (ANN), deep NN (DNN), and convolutional NN (CNN) are the mainstream to further eliminate nonlinear distortions of PAM4 optical links with higher complexity. When the SVM is applied to PAM4 signal, sensitivity gain of 2.5 dB is obtained for 60 Gbit/s VCSEL-MMF short-reach optical links [40]. Yang et al. experimentally demonstrate a C-band 4 × 50 Gbit/s PAM4 transmission over 80 km SSMF employing a radial basis function ANN [44]. In [46], with the help of CNN, a 112 Gbit/s PAM4 transmission over 40 km SSMF is accomplished and the BER performance outperforms traditional VNLE. For more complicated DNN, the BER of $4.41 \times 10^{-5}$ is obtained for a 64 Gbit/s PAM4 transmission over 4 km MMF based on 850 nm VCSEL [47]. Note that some equalization methods such as look-up table (LUT) are not mentioned in this paper due to less implementation.

**Table 3.** Equalization technologies for short-reach PAM4 optical links.

| Equalization | Distortion | Rate (Gbps) | Reach (km) | Wavelength (nm) | FEC | Tx | Rx | Ref. |
|---|---|---|---|---|---|---|---|---|
| FFE/DFE | bandwidth limitation & CD | 56 | 60 | 1295.13 | $3.8 \times 10^{-3}$ | EML | APD + TIA & PD + TIA | [26] |
| FFE/DFE | bandwidth limitation & CD | 106 | 40 | 1309.49 | $2 \times 10^{-4}$ | EML | APD | [27] |
| FFE/DFE | bandwidth limitation & CD | 112 | 40 | 1314 | $1 \times 10^{-3}$ | EML | PD + TIA | [28] |
| THP | bandwidth limitation & power fading | 100 | 40 | 1300 | $4 \times 10^{-3}$ & $2 \times 10^{-4}$ | EML | PD + TIA | [72] |
| DD-FTN | enhanced noise | 140 | 20 | 1296.2 | $3.8 \times 10^{-3}$ | EML | PD + TIA | [35] |
| ID-FFE/ ID-DFE | chirp of modulator | 56 | 43 | N/A | $3.8 \times 10^{-3}$ | DML | PD | [37] |
| CR-FFE | clock offset with CD | 50 | 40 | 1549.39 | $2 \times 10^{-2}$ | DML | OA + PD | [39] |
| VNLE | nonlinearity | 56 | 100 | 1549.81 | $3.8 \times 10^{-3}$ | DML | OA + PD | [92] |
| VNLE | nonlinearity | 56 | 70 | 1311.89 | $3.8 \times 10^{-3}$ | DML | OA + PD | [93] |
| SVM | nonlinearity of VSCEL | 60 | 0.05 | 850 | $3.8 \times 10^{-3}$ | VSCEL | PD | [40] |
| SVM | level-dependent skew | 50 | 20 | 1551 | $1 \times 10^{-3}$ & $2 \times 10^{-4}$ | DML | PD | [41] |
| ANN | nonlinearity of DML | 20 | 18 | 1310 | $3.8 \times 10^{-3}$ | DML | PD | [43] |
| ANN | nonlinearity | 50 | 80 | 1551.35 | $3.8 \times 10^{-3}$ | DML | OA + PD | [44] |
| CNN | nonlinearity | 56 | 25 | N/A | $3.8 \times 10^{-3}$ | DML | APD | [45] |
| CNN | nonlinearity | 112 | 40 | 1293 | $2 \times 10^{-4}$ | EML | PD + TIA | [46] |
| DNN | nonlinearity | 64 | 4 | 850 | $2 \times 10^{-4}$ | VSCEL | PD + TIA | [47] |
| DNN | nonlinearity | 50 | 80 | 1551.35 | $1 \times 10^{-3}$ | DML | OA + PD | [48] |

At present, the standard to evaluate the equalization technologies is considering both the performance improvement and the computational complexity. The high complexity will occupy massive resources for computation, thus increasing the computational cost. The score to judge the equalization technologies can be described as:

$$\text{score} = \frac{a * \text{performance improvement}}{b * \text{computational cost}} \tag{7}$$

where *a* and *b* are the influence factors of performance improvement and computational cost, respectively. The comparison of the computational complexity and performance improvement for different equalization technologies is summarized in Table 4. Note that the main limited factor for the short-reach PAM4 system is the cost, and the goal for equalization technologies is to achieve better performance at the same computation cost. As is shown in Table 4, the SVM an NN have a higher complexity than conventional equalization, although the performance can be significantly improved. However, the computation cost is getting lower as the continuous development of the integrated circuit. So, from the long-term point of view, the equalization based on machine learning algorithms will play an increasingly important role for short-reach PAM4 optical links.

**Table 4.** Comparison of the computational complexity and performance improvement for different equalization technologies.

| Equalization Technologies | FFE/DFE | DD-FTN | ID-FFE/ ID-DFE | CR-FFE | VNLE | Sparse VNLE | SVM | NN |
|---|---|---|---|---|---|---|---|---|
| Computational Complexity | low | fair | low | low | high | fair | high | high |
| Performance Improvement | low | fair | fair | fair | high | high | high | high |

## 5. Conclusions and Perspective

In this paper, we have reviewed various equalization technologies for short-reach PAM4 optical links. A typical system configuration is presented and the comparisons among different transmitters and receivers are introduced. Different distortions including linear impairments, device nonlinearity and power fading effect are described, which are induced by low-cost components and need to be mitigated by pre- and post-equalization. The conventional equalizers, including FFE and DFE, are illustrated to eliminate the linear impairments such as bandwidth limitation and chromatic dispersion, while the nonlinear distortions are compensated for by conventional VNLE and sparse VNLE. The machine learning algorithms like SVM and NN are proposed to further mitigate severe nonlinear distortion and achieve significant performance improvement. Finally, a summary is given for different equalization technologies and a standard for the evaluation of equalizers is defined to consider both performance improvement and computational complexity. Additionally, as the cost of computation constantly decreases, equalization technologies based on machine learning may become the mainstream technology for next-generation short-reach PAM4 transmission.

**Author Contributions:** This paper was mainly wrote by H.Z. and Y.L. (Yan Li) Y.L. (Yuyang Liu) provided the idea and J.W. supervised overall project. L.Y., C.G., W.L., J.Q., H.G., X.H., Y.Z. and J.W. contributed to the reviewing and editing of the manuscript.

**Funding:** This paper is partly funded by National Natural Science Foundation of China (NSFC) (61875019, 61675034, 61875020, 61571067); The Fund of State Key Laboratory of IPOC (BUPT); The Fundamental Research Funds for the Central Universities.

**Acknowledgments:** We would like to thank the unknown reviewers that helped improve this manuscript with their valuable feedback.

**Conflicts of Interest:** The authors declare no conflict of interest.

## Abbreviations

| | Acronyms | | Acronyms |
|---|---|---|---|
| ADC | Analog-to-digital converter | APD | Avalanche photodiode |
| BER | Bit error rate | CAP | Carrier-less amplitude and phase modulation |
| CBT | Complete binary tree | CD | Chromatic dispersion |
| CR | Clock recovery | DAC | Digital-to-analog converter |
| DCI | Data center interconnects | DD | Direct detection |
| DFE | Decision feedback equalizer | DML | Directly modulated laser |
| DMT | Discrete multi-tone | DSP | Digital signal processing |
| EML | Electro-absorption modulated lasers | FEC | Forward error correction |
| FFE | Feed-forward equalizer | FTN | Faster than nyquist |
| HD | Hard decision | ID | Intensity directed |
| IM | Intensity modulation | ISI | Inter-symbol interference |
| KK | Kramers-Kronig | LMS | Least mean squares |
| MLSE | Maximum likelihood sequence estimation | ML | Machine learning |
| MMF | Multi-mode fiber | MZM | Mach-Zehnder modulator |
| NN | Neural network | OA | Optical amplifier |
| OPBF | Optical band-pass filter | OSNR | Optical signal-to-noise ratio |
| PAM4 | Four-level pulse amplitude modulation | PD | Photodiodes |
| PRBS | Pseudorandom binary sequence | QAM | quadrature amplitude modulation |
| ReLU | Rectified Linear Units | RLS | Recursive least squares |
| SCO | Sampling clock offset | SMMF | Standard single-mode fiber |
| SMO | Sequential minimal optimization | SPM | Self-phase modulation |
| SVM | Support vector machine | TIA | Trans-impedance amplifiers |
| VCSEL | Vertical cavity surface-emitting laser | VNLE | Volterra nonlinear equalizer |
| VR | virtual reality | ZF | Zero-forcing |

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
