# Peer review of "Recent Advances in Equalization Technologies for Short-Reach Optical Links Based on PAM4 Modulation: A Review"

_applsci, doi:10.3390/app9112342_

Round 1
Reviewer 1 Report
The paper "Recent Advances in Equalization Technologies for Short-Reach Optical Links Based on PAM4 Modulation: A Review" deals with the interesting topic of equalization technologies for short-reach optical transmissions. The paper is generally clear and well-written. In my opinion the paper should mention also other equalization techniques, e.g., zero-forcing equalization (see, for example, "High-speed PAM4-based optical SDM interconnects with directly modulated long-wavelength VCSEL"). These linear equalization techniques are well-known in the context of wireless communications and used to face the problem of mitigating narrowband-interference for example (see "A constrained maximum-SINR NBI-resistant receiver for OFDM systems" or "NBI-resistant zero-forcing equalizers for OFDM systems" and references therein). Finally, in order to improve the readability of the paper I suggest to add a table with the main acronyms utilized in the paper.
Reviewer 2 Report
This paper provides a good overview of equalization technologies for optical links. The work provides a good survey of the problems in this area and comparing and contrasting approaches in the literature. This work is valuable as a reference but could use a bit of work on the grammar before final submission.
Reviewer 3 Report
A review article should take stock of the situation and present the evolution of the topic. The authors in this manuscript achieve this aim by presenting an extensive list of references, including the most recent. However I present some suggestions for improvement:
· A table with the different techniques and results helped to a better understanding evolution for PAM4 optical links (like table3). Advantage ? Disadvantage? The system in figure 1 is the typical? One reference is necessary. In table 1 and 2 is necessary references. Ireccommend extent section 2 whre the author need explain each compoment and the possibility of gains in quality.
· In section 3 change the title. In sub -section 3.1, the authors intend to present a theoretical section? Is a bit confused. The authors have not yet said where they could change to increase the quallity of the system. Filters? Fonts? I reccomend rewrite this section. Each figure needs a reference. The section 3.2 present 2 problem in the gain of quality. Others is necessary? Or these are the most important?
· In section 4 the authors propose describe the techniques for resolve some problema im PAM4. I recommned a table with techniques used and the most important. In the different sub-section the author describe some technique. These are the most importante? Reference for equation and figures are necessary.
· In the different section the authors presente equations. Is necessary a references. A good theoretical section can be introduce. This section can improve the manuscript.
· For each technique in table 3 for equalization technologies for short-reach PAM4 optical links i recommend the authors present a short description and a most important results.
· Clarify the equation 6. Some parameter are confuse.
· The section 5 is poor. I recommend rewrite.
I recommend accept this manuscript to publish after one minor review.
Round 2
Reviewer 1 Report
This revised version can e accepted for publication.